# Compound-Specific ^14^N/^15^N Analysis of Amino Acid Trimethylsilylated Derivatives from Plant Seed Proteins

**DOI:** 10.3390/ijms23094893

**Published:** 2022-04-28

**Authors:** Jean-Baptiste Domergue, Julie Lalande, Cyril Abadie, Guillaume Tcherkez

**Affiliations:** 1Institut de Recherche en Horticulture et Semences, Université d’Angers, INRAe, 42 Rue Georges Morel, 49070 Beaucouzé, France; jean-baptiste@domergue.net (J.-B.D.); julie.lalande@inrae.fr (J.L.); cyril.abadie@inrae.fr (C.A.); 2Research School of Biology, Australian National University, Canberra, ACT 2601, Australia

**Keywords:** stable isotopes, natural abundance, gas chromatography, silylation, exact mass spectrometry, nitrogen-15

## Abstract

Isotopic analyses of plant samples are now of considerable importance for food certification and plant physiology. In fact, the natural nitrogen isotope composition (δ^15^N) is extremely useful to examine metabolic pathways of N nutrition involving isotope fractionations. However, δ^15^N analysis of amino acids is not straightforward and involves specific derivatization procedures to yield volatile derivatives that can be analysed by gas chromatography coupled to isotope ratio mass spectrometry (GC-C-IRMS). Derivatizations other than trimethylsilylation are commonly used since they are believed to be more reliable and accurate. Their major drawback is that they are not associated with metabolite databases allowing identification of derivatives and by-products. Here, we revisit the potential of trimethylsilylated derivatives via concurrent analysis of δ^15^N and exact mass GC-MS of plant seed protein samples, allowing facile identification of derivatives using a database used for metabolomics. When multiple silylated derivatives of several amino acids are accounted for, there is a good agreement between theoretical and observed N mole fractions, and δ^15^N values are satisfactory, with little fractionation during derivatization. Overall, this technique may be suitable for compound-specific δ^15^N analysis, with pros and cons.

## 1. Introduction

Intense efforts are currently devoted to find markers of seed quality, including indices of nutritional value (for a recent example in Legumes, see [1]). In particular, the nitrogen content and isotope signature can be useful to characterize seed content in protein and the nitrogen source [2,3,4]. Unlike other crops, legumes produce protein-rich seeds (up to 40% in lupine) and are capable of atmospheric N_2_ fixation, leading to a specific ^15^N natural abundance, close to 0‰. In cereals such as wheat, both the content and physical property of seed protein fraction are of importance for food quality, and the δ^15^N value essentially depends on nitrogen nutrition via fertilisation [5]. Further information can be extracted using compound-specific isotopic analysis of amino acids from seed protein, not only to trace back the isotope composition of essential amino acids in human nutrition, but also to follow metabolic modification in plants when environmental conditions vary under climate change. For example, it has been shown that wheat grain protein shows alteration in both free amino acids and protein composition under elevated CO_2_ [6,7] and in principle, this effect could change amino acid δ^15^N values. This, in turn, may influence δ^15^N values in human diet.

Amino acid δ^15^N analysis can be performed using gas chromatography/combustion/isotope ratio mass spectrometry (GC-C-IRMS). Many studies took advantage of this technique (reviewed in [8,9]). GC-C-IRMS technology couples GC separation of volatile compounds to isotopic analysis by IRMS via a combustion module which converts compounds to N_2_ (the IRMS, thus, monitors signals of mass-28, 29 and 30 of N_2_ isotopologues). Several derivatization methods have been developed to convert amino acids to volatile derivatives (reviewed in [9]). Amongst the most successful methods is “pivaloylation”; that is, the conversion of amino acids into their *N*-pivaloyl-*i*-propyl esters. This method is efficient in that it gives good chromatographic resolution and stable derivatives [10,11,12]. However, it is not adapted to all amino acids (a maximum of 15 amino acids can be analysed), for instance Gln and Glu (indistinguishable derivative) or Arg, which is represented in high amount in plant seed proteins. Alternative derivatization methods specific to Arg do exist, such as PFPA (pentafluoropropionic anhydride) utilisation, which converts Arg into a fluorinated derivative [13]. Although efficient in keeping the Arg backbone intact, this technique is problematic due to fluorinated products such as HF (hydrofluoric acid) that can form in the reactor. Taken as a whole, there are pros and cons associated with these techniques, making them useful for some amino acids but not for others.

Importantly, the major drawback of current techniques is that the identification of products other than classical, expected amino acids derivatives requires tedious chemical analysis and analytical standards. By contrast, trimethylsilyl derivatization, often considered as unsuitable for isotopic analyses, is *(i)* routinely used in metabolomics, with well-established protocols, and *(ii)* associated with publicly available databases, allowing facile identification (NIST, and Golm Metabolomics Database). Therefore, the question of the potential of trimethylsilylation for δ^15^N analysis of amino acids must be asked. It has been recognised that this derivatization causes problems with carbon, not only due to silicium carbide formation in the reactor but, also, selective loss of C atoms and the need of numerically important isotopic corrections (such problems are reviewed in [9]). For δ^15^N analysis, a closely related method, MTBSTFA (*N*-tert-butyldimethylsilyl-*N*-methyltrifluoroacetamide) based silylation, has proven useful for amino acid isotopic analysis, including in plants [14,15,16,17,18,19]. Similar to other non-silylated reagents, the major problem is the lack of databases for compound identification.

Here, we used trimethylsilylation (like in metabolomics) and carried out isotopic analysis of amino acids from plant seed protein hydrolysis. Our objective was to assess whether trimethylsilylation was suitable for amino acid δ^15^N analysis by GC-C-IRMS and identify potential problems that may occur with this method. GC-MS analyses were carried out with exact mass spectrometry to allow non-ambiguous identification of products. The use of standards allowed use to assess whether this method could provide acceptable estimates of δ^15^N values. We also looked at possible isotope effects when multiple derivatives were formed. Since we carried out δ^15^N analyses, we removed methoxyamine from the usual derivatization mixture used in metabolomics and used MSTFA (trimethylsilylation reagent) alone in pyridine. This modification allowed us to focus on amino acids without interference with N-containing methoxime derivatives of sugars.

## 2. Results

### 2.1. Chromatograms and Observed Peaks

Samples were analysed not only by GC-C-IRMS but also by GC-MS in order to identify peaks. As expected, most peaks were associated with amino acid TMS derivatives (some sugar impurities being found occasionally in GC-MS only) (Figure 1). Nearly all amino acids were detected, with the exception of Trp, Cys (in Legume proteins) and Met, which were below the detection limit in GC-C-IRMS and showed a very small peak in GC-MS. Several amino acids had a very close retention time, leading potentially to peak shouldering in GC-C-IRMS, such as Ile and Pro at about 7.1 min, and ornithine, Glu and Phe at about 9.9 min. Several amino acids formed several derivatives: Gly (Gly 2TMS and 3TMS), Pro (Pro 1TMS and 2TMS) and Asp (Asp 2TMS and 3TMS). Arg yielded several derivatives: citrulline 3TMS, ornithine 3TMS and ornithine lactam 3TMS (products of Arg derivatization are addressed in detail below). It is worth noting that the signal in total ion current (TIC) on the GC-MS was not correlated to the number of moles of the derivative of interest, as shown by the comparison of GC-MS and IRMS traces in Figure 1. This effect simply comes from the variable response coefficient (ionization efficiency) in the source of the GC-MS by electron impact. This effect is very well-known and explains why GC-MS analyses are commonly used for semi-quantitative analysis rather than absolute quantitation. Here, the IRMS trace gives direct access to N atom quantitation since the intensity on collectors is directly proportional to the amount of N_2_ molecules (generated in the combustion tube).

### 2.2. N Mole Fraction

The signal obtained by GC-C-IRMS with the mass-28 isotopologue was used to compute a relative signal (contribution to total) and compare with the theoretical N mole fraction deduced from the amino acid sequence of seed proteins (retrieved from Uniprot; see Section 4 for more details). Such a comparison is shown in Figure 2. There was a generally good agreement between observed and expected N mole fraction. In particular, Gln prevailed in gliadin, while protein from Legumes were enriched in Arg. Several amino acids were either not detected or associated with a very weak signal, e.g., Cys and Trp. In all cases, there was an overestimation of the Gly N mole fraction, while in Legumes Arg was underestimated and in wheat gliadin, Gln was underestimated. The origin of the overestimation of Gly N mole fraction is unknown. It is possible that during hydrolysis, some unforeseen amino acid cleavage generated Gly. In the case of Arg, this simply comes from the production of several derivatives, with probable loss of N despite the correction made to account for this effect (see also below; Figure 4). The underestimation of Gln in gliadin is also not surprising because of the conversion to glutamate and pyroglutamate. In fact, conversely, Glu is overestimated by the same amount as the gap between observed and expected Gln (Figure 2a). Taken as a whole, in a reciprocal plot (Figure 2b), all data points were close to the 1:1 relationship (well below 10% error), except for Glu and Gln (in gliadin) and Arg (in Legumes).

### 2.3. Potential Kinetic Isotope Effects in Derivatization

Despite the fact that silylation is fast and complete (with an excess of silylation reagent), isotope fractionations can occur when multiple products are formed. In fact, the isotope composition of the different derivatives could be different, while the weighted δ^15^N average remains equal to the original, non-derivatized metabolite. We examined this problem using Thr (which can give either Thr 2TMS or Thr 3TMS) and Glu (which can give either Glu 3TMS or pyroglutamic acid 2TMS) (Figure 3a). Different amounts of starting material were used to vary proportions of derivatives. Regardless of proportions, there was a small isotopic offset between derivatives of roughly 2‰ (3TMS derivatives ^15^N-enriched by 2 per mil compared to their counterparts) (Figure 3b). When plotted in a log–log representation, the apparent ^14^N/^15^N isotope effect associated with “pyroglutamatization” and TMS loss from Thr 3TMS was found to be about 1.0025 and 1.001 (Figure 3c,d), respectively. The term “apparent” is used here because, in reality, pyroglutamic acid 2TMS is not formed from Glu 3TMS but is formed concurrently. Additionally, Thr 2TMS is not formed from Thr 3TMS but is generated concurrently during derivatization. These isotope effects are small and suggest that the isotopic difference between derivatives and their average is within 1‰.

### 2.4. Silylated Derivatives of Arginine

Arginine is very well-known to be transformed to compounds other than Arg 3TMS during silylation and, in particular, to form citrulline 3TMS. This is an important issue because proteins from Legume seeds are rich in Arg (about 30% of protein total N, Figure 2). Here, we checked the products of derivatization, because under our conditions, there was no methoxyamine (unlike in protocols used routinely for metabolomics). In principle, upon derivatization, Arg can form four products: Arg 3TMS, citrulline 3TMS (via the loss of one N atom) and ornithine (via the loss of the guanidium group), which can, in turn, generate two products: ornithine 3TMS and ornithine lactam 2TMS (Figure 4a). When derivatized and injected pure, Arg essentially formed ornithine 3TMS and ornithine lactam 2TMS, citrulline 3TMS being nearly negligible (Figure 4b, mass-28 trace in red). This is in clear contrast with GC-MS signals because ionisation efficiency of citrulline 3TMS is high and, thus, generated a signal as high as ornithine (Figure 4b, TIC trace in black). Interestingly, Arg 3TMS is negligible under such conditions. By contrast, it is significant when methoxyamine is added. Its retention time is very close to that of citrulline 3TMS (11.41 min). However, exact mass MS analysis clearly shows the appearance of fragments typical of Arg 3TMS, such as ion *m/z* at 187.10776, that are absent in citrulline 3TMS fragmentation (Appendix A).

### 2.5. δ^15^N Values and Performance of the Method

The efficacy of the method was first examined using pure standards. We chose Arg, Glu and Thr due to their multiple derivatives, and norleucine as a common standard used in GC-C-IRMS adapted to amino acids (Table 1, top). The δ^15^N value indicated for Arg, Glu and Thr is the weighted average of the derivatives. In practice, the δ^15^N value of derivatives is rather close (see Section 2.3 for Thr and Glu) including for Arg: we found that ornithine lactam 2TMS, ornithine 3TMS and citrulline 3TMS were at −5.79 ± 0.08, −5.23 ± 0.72, and −4.91 ± 1.27, respectively. Taken as a whole, there was a rather good agreement between values obtained by GC-C-IRMS and EA-IRMS. The maximum error in accuracy was for norleucine, of 0.83‰.

There were quite significant variations amongst amino acids in plant seed protein hydrolysates (Table 1, middle). Despite such variations, there were similarities between lupine and alfalfa, for example, with naturally ^15^N-enriched Thr and Phe, and ^15^N-depleted Ala. In gliadin hydrolysate, Ala generated a signal that was too weak to allow reliable δ^15^N determination. In addition, Ile 2TMS, Pro 2TMS and Gly 3TMS were too close to allow proper integration for δ^15^N measurement, although peak shouldering was sufficiently apparent to estimate their relative content (Figure 2). To assess whether δ^15^N values in amino acids were reliable, we computed their weighted average (i.e., weighted with observed N fraction) and compared with the value obtained by EA-IRMS on non-hydrolysed protein isolates (Table 1, bottom). There was a relatively good agreement between them, within 0.4 to 1.4‰. It must be kept in mind that this estimation by computation from individual values of amino acids was an approximation because for some of them, the signal was too weak to determine the isotope composition (we imposed the limit of 0.4 nA, under which linearity is lost). The repeatability of δ^15^N values was generally good, with often less than 1.5‰ uncertainty. It was occasionally found to be higher, in particular in gliadin amino acids. The reason for the accrued uncertainty for gliadin is presently unknown. For example, it could be that the gliadin powder, which tends to form lumps, was isotopically heterogenous.

## 3. Discussion

### 3.1. Pros and Cons of the Silylation Method for δ^15^N Analysis by GC-C-IRMS

As stated above in the Section 1, trimethylsilylation is not a commonly used derivatization method for δ^15^N analysis [8]. Reasons includes derivative instability and multiplicity, silicium deposition in the combustion reactor (in the form of silicium carbide or similar compounds), and sometimes insufficient chromatographic resolution. Here, we had to adapt the protocol used for metabolomics by withdrawing methoxyamine to prevent the synthesis of N-containing products that had nothing to do with amino acids (e.g., glucose derivatives that would come from starch impurities during protein preparation).

Several amino acids yielded multiple derivatives, but they were found to have rather similar δ^15^N values; therefore, their isotopic impact appears to be modest. We also recognize that derivation products leading to N loss and cleavage can cause an isotopic fractionation [20] if *(i)* there is an isotope effect during the reaction; *(ii)* there is a high difference in δ^15^N values amongst N atoms of the same molecule (for example between amino N and guanidium N in Arg). Under our conditions, we found little isotopic fractionation (up to 2.5‰) suggesting that effect *(i)* is not strong. Effect *(ii)* is presently difficult to assess since to our knowledge, intramolecular δ^15^N analyses are limited to few examples [21,22]. Chromatographic resolution was occasionally an issue, since it was not possible to integrate peaks separately and thus obtain reliable δ^15^N values in, e.g., Ile, Pro and Gly in gliadin (Figure 1, Table 1). Nevertheless, it should be possible in the future to use deconvolution algorithms applied separately to mass-28, mass-29 and mass-30 to extract isotopic signals and obtain δ^15^N values even when peaks partly overlap.

Despite these disadvantages, our results show that the trimethylsilylation method can be useful. First, its clear advantage is the use of databases from GC-MS analyses allowing facile product identification (including amino acid derivatives; illustrated in Figure 1). Second, implementation of derivatization steps is easy and rapid (MSTFA in pyridine), and silylation reagents are cheap. Third, both amino acid standards and protein isolates show that accuracy is reasonable (Table 1). Furthermore, we note that our δ^15^N values are very close to that found in [15] in wheat, and show similar δ^15^N differences (with respect to Glu) in wheat and Legumes to that found in [18] (with the exception of Gly and Ser in Legumes, further discussed below).

### 3.2. Biological Significance of Amino Acid δ^15^N

The metabolic origin of δ^15^N in amino acids has been studied previously either in seed protein or leaf free amino acids [17,18,23]. Here, amino acids were obtained from protein hydrolysis and are thus representative of nitrogen allocation from leaves to seeds during seed and fruit development. The δ^15^N value in protein isolates was found to be close to 0–1‰ in both Legume species (lupine and alfalfa), reflecting N assimilation from symbiotic atmospheric N_2_ fixation [24,25,26]. By contrast, gliadin was found to be slightly enriched (+3.53‰ on average), reflecting the use of ^15^N-enriched nitrate coming from fertilizers. Differences amongst amino acids are the consequence of isotope fractionations during enzymatic reactions. For example, Phe is ^15^N-enriched by 6–7‰ (compared to Glu) in all three species, due to the isotope fractionation in Phe ammonia lyase which discriminates against ^15^N by ≈ 2‰ (for a recent work on this aspect, see [27]). Gly was found to be ^15^N-depleted compared to Ser likely due to the inverse kinetic isotope effect in glycine decarboxylase-serine hydroxymethyl transferase, which liberates ^15^N-enriched ammonia and leaves behind Gly depleted in ^15^N [23]. As well, Gly can be synthesized from Thr cleavage (see below). 

Ornithine lactam (from Arg degradation during derivatization) appeared to be variable, being either enriched (gliadin) or depleted (alfalfa) in ^15^N. The original isotope composition in Arg is determined by both the δ^15^N of N sources (Glu, and ammonia via carbamoyl-phosphate) and isotope fractionation in Arg synthesis and degradation. Carbamoyl phosphate synthase is believed to fractionate by about 11‰ [28] while argininosuccinate lyase and arginase fractionate by −4 and +10‰, respectively [29,30,31]. Therefore, depending on the δ^15^N value in free ammonia and the metabolic flux in Arg turn-over, some variations in δ^15^N of Arg are expected. We also note that all branched chained amino acids (Val, Leu and Ile) appear to be ^15^N-depleted (by up to *c*. 7‰ compared to Glu), demonstrating an isotope effect in branched chain amino acid amino transferase (as suggested by [32]).

Interestingly, Thr was found to be considerably ^15^N-enriched in protein from Legumes and not in gliadin. This difference reflects the fundamental difference in Gly, Thr and Ile balance between wheat and Legumes. In wheat, in phloem sap feeding developing grains, Gly, Thr and Ile concentration are within the same order of magnitude (1–2% of total amino acids) [33,34]. In Legumes, Ile and Thr are present, but Gly is only in trace amount [35,36]. In Legume developing seed tissues, Gly is thus reformed via Thr cleavage by Thr aldolase. In fact, this enzyme has been shown to be critical for Legume seed protein content and is believed to be important to control seed Thr levels [37,38]. Thr aldolase involves a Schiff-base intermediate in the mechanism, and likely fractionates between N isotopes, causing a ^15^N-enrichment in Thr left behind. This effect is expected to be visible in Legumes in which developing seeds synthesize Gly from Thr. It is worth noting that this situation is in clear contrast with leaf amino acids, where Thr can occasionally be ^15^N-depleted [14,17]—a phenomenon also found in animals [16] and referred to as the “Thr anomaly” [39]. This phenomenon must be explained by normal kinetic isotope effects during Thr synthesis rather than an inverse isotope effect in Thr degradation, since Thr deaminase (used to synthesise Ile) is associated with a normal isotope effect [39]. In plant leaves, a simple possibility is the involvement of cystathionine gamma synthase. This enzyme consumes the direct precursor of Thr synthesis, *O*-phospho homoserine. It involves a Schiff base in catalysis and thus certainly fractionates between N isotopes.

## 4. Materials and Methods

### 4.1. Source Materials

Chemicals (pyridine, MSTFA) were from Sigma-Aldrich (Merck, Saint Quentin Fallavier, France), as well as purified gliadin, which is available commercially in the form of a powder (reference G3375). Lupin (*Lupinus albus*) flour was purchased from Priméal (batch no. 619746, 2021). Barrel alfalfa (*Medicago truncatula*) seeds were obtained from INRAe Angers greenhouses (cultivated in 2019) from cultivar no. DZA-315-16 M14 belonging to the *Medicago truncatula* core collection. Greenhouse air was constantly renewed (with temperature and humidity automated control) so that the δ^15^N value of N_2_ in air was 0‰. Isotopic standards used in GC-C-IRMS were purchased from Sigma-Aldrich (caffeine batch no. BCCC8969, 2021, and nicotine batch no. BCCD0109, 2021) and their δ^15^N was determined by EA-IRMS using caffeine (IAEA-600, batch no. 860) as a reference material.

### 4.2. GC-MS Analyses

Gas chromatography/mass spectrometry (GC-MS) analyses were carried out using a GC-MS-Orbitrap Q Exactive (Thermo Scientific, Courtaboeuf-Les Ulis, France). Fifteen µL of protein hydrolysis extract were poured into a vial (with insert) and spin-dried at 39 °C. Samples were derivatized (automatically with a preparative robot) with or without 20 µL methoxyamine (20 mg mL^−1^ in pyridine; 90 min at 37 °C) and 30 µL N-methyl-N(trimethylsilyl)trifluoroacetamide (MSTFA) for 30 min at 37 °C. Prior to injection, 5 µL of alkane mix (14 alkanes from C_9_ to C_36_, 3 µg µL^−1^, Connecticut n-Hydrocarbon Mix, Supelco) were added in each sample to compute the retention index. Analyses were performed by injecting 1 µL in splitless mode at 230 °C (injector temperature) in a TG-5 SILMS column (30 m × 0.25 mm × 0.25 µm; Thermo Scientific) set in a Trace 1300 Series GC (Thermo Scientific). Helium was used as gas carrier with a constant flow of 1 mL min^−1^. After one minute at initial GC oven temperature (70 °C), temperature was raised to 325 °C at 15 °C min^−1^ and finally kept at 325 °C for 4 min. MS analyses were operated in positive polarity in full MS scan mode with the following source settings: mass scan range 50–750 *m/z*, resolution 60,000, AGC target 1E6, MS transfer line 300 °C and filament delay 4.12 min. Ionisation by electron impact (70 eV) was performed at 250 °C ion source temperature. Amino acids were identified automatically using TraceFinder (Thermo Scientific) using retention time, major characteristic fragment (*m/z* ion) and a confirmation fragment, with a maximum tolerance of 0.00007 Da. 

### 4.3. GC-C-IRMS Analyses

Gas chromatography/combustion/isotope ratio mass spectrometry (GC-C-IRMS) was carried out with a GC Agilent 7890B coupled to an autosampler Agilent 7693A, GC5 combustion module (Elementar, Langenselbold, Germany), and an IRMS Isoprime precisION monitored with the software ionOs (Elementar). The combustion module was filled with the GCN reactor as the furnace tube. The tube was re-oxidized with O_2_ (with the method GC-O_2_-Flush for 4 h every 80 samples). CO_2_ was cryogenically trapped with liquid N_2_. Source parameters were optimized for N_2_ and analyses were run at 600 µA trap current. Before sample injection in ^15^N mode, analyses in ^13^C mode were carried out with nicotine and caffeine (in methanol) as well as the alkane mix (same as for GC-MS) to compute retention indices (used for identification). The GC sequence was the same as for GC-MS while the valve (switch from “Waste” to “Flow to IRMS”) was opened at 6.15 min to avoid entry of solvent and reagents into the IRMS. The injection volume was 1 µL with a split ratio of 2. The GC column was similar to that of the GC-MS (HP-5, 30 m × 0.32 mm × 0.25 µm). Two reference gas (N_2_) injections were done before opening and after closure of the valve. Reference N_2_ gas (δ^15^N = −0.68‰) pressure was set to have a square signal at 7 nA for mass-28. To avoid complication in the chromatogram with an internal standard, we used external isotopic standards (nicotine and caffeine) injected two times before and after each batch of 3 to 4 samples. EA-IRMS analyses were carried out using a Pyrocube (Elementar) coupled to the IRMS Isoprime precision, with samples weighted in tin capsules.

### 4.4. N Mole Fraction

Theoretical amino acid content in proteins was calculated from the sequence of conglutins (lupine), vicilins, legumins and albumins (alfalfa) and gliadins (wheat), retrieved from Uniprot (www.uniprot.org, accessed 15 March 2022). The content was expressed in % of total nitrogen thus took into account the number of N per amino acid. Experimental mole fractions were computed using the height of the mass-28 peak of amino acid on the GC-C-IRMS trace and expressed in % (relative to total). Nearly identical results were obtained with areas instead of height for amino acid that were sufficiently resolved in the chromatogram. A correction was implemented for Arg, where the observed peaks of ornithine derivatives were multiplied by two to account for the fact that Arg contains 4 N atoms. The peak of pyroglutamic acid 2TMS was used to estimate the Gln content. Asn was degraded to Asp; thus, the N mole fraction chart in Figure 2 mentions Asx and does not distinguish Asp and Asn. Pro was estimated using the sum of peaks associated with Pro 1TMS and Pro 2TMS. Similarly, Lys was estimated using the sum of its derivatives.

### 4.5. Protein Extraction and Hydrolysis

Gliadin was used directly from the commercial powder. Alfalfa and lupine seed proteins were extracted from 20 mg powder with 2 mL NaCl 5% and precipitated with 6 mL methanol. After centrifugation at 10,000× *g* for 10 min, the supernatant was discarded, and the protein isolate was dried. Hydrolysis was carried out in Ace pressure tubes with HCl 6 M at 110 °C for 16 h. The hydrolysate was dried under N_2_ flow, resuspended in 1 mL methanol:water (50:50 v:v), centrifuged and the supernatant was transferred into a 2 mL Eppendorf. Forty (40) µL were transferred to a vial (with insert) for GC-MS and spin-dried, the rest being spin-dried directly in the Eppendorf tube. Derivatization was done manually in the same tube by adding MSTFA (30 µL) in pyridine (20 µL), with 30 min (37 °C) incubation time. The derivatized extract was transferred into a vial for GC-C-IRMS.

### 4.6. Log-Log Plots and Isotope Effects

Isotope effects are denoted as α and defined as ^14^N/^15^N ratio of rates. Therefore, the ^15^N/^14^N isotope ratio (*R*) of the substrate of a reaction associated with an isotope effect α is given by (Rayleigh’s equation): ln(*R*) = ln(*R*_0_) + (1/α − 1)⋅ln(1 − *f*) where *R*_0_ is the initial isotope ratio (when the reaction starts) and *f* is the mole fraction of substrate that has been converted to product (1 − *f* is thus the fraction of remaining, unreacted substrate). Thus, when ln(*R*) − ln(*R*_0_) is plotted against ln(1 − *f*), the slope is (1/α − 1) and thereby gives α.

## 5. Conclusions and Perspectives

Taken as a whole, our results show that isotopic analysis by GC-C-IRMS via trimethylsilylation as a derivatization method is feasible for amino acids from seed proteins. Amongst disadvantages encountered with this method, resolution is probably the most critical since isotope effects in fragmentation and derivatization are limited. Future studies are thus warranted to develop deconvolution tools to extract individual signals when there is some coelution. In principle, it must be easily implementable using signal fitting such as beta-type distribution (rather than gaussian bell-shaped curve fitting) to account for the slightly non-symmetrical shape of peaks in GC-C-IRMS (e.g., when the reactor ages). This would allow the utilization of common protocols for both metabolomics and isotopomics and, thus, should facilitate metabolic studies. This aspect will be addressed in a subsequent study.

## Figures and Tables

**Figure 1 ijms-23-04893-f001:**
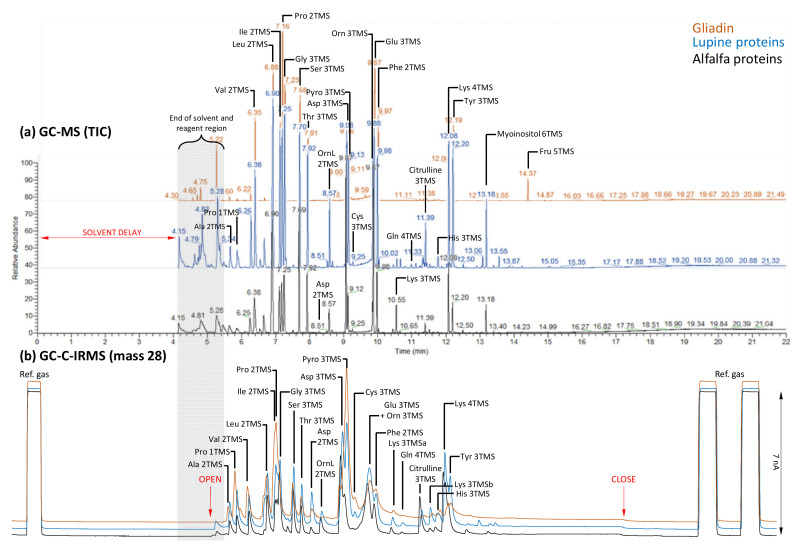
**GC-MS (total ion current, TIC)** (**a**) **and IRMS (mass-28)** (**b**) **traces of protein hydrolysis samples**. This figure shows typical traces for the three sample types, graphically separated by an offset (*y*-axis) to facilitate reading. Peaks are labelled with names showing the number of TMS (trimethylsilyl) groups. Note the presence of sugar impurities in samples (myoinositol, fructose). The region corresponding the end of the solvent (and reagents) elution time window is shaded in grey (left). In (**b**), time has been readjusted (thanks to alkanes) to show the coincidence between TIC and mass-28 traces (peaks) and, thus, correct for the longer retention time in GC-C-IRMS analyses due to the preparative line. Letters “a” and “b” for Lys 3TMS refer to two possible forms of Lys 3TMS (one or two N atoms carrying TMS groups). “Pyro” and “OrnL” stand for pyroglutamate (Figure 3) and ornithine lactam (Figure 4), respectively. The flow to IRMS (switch from waste to analysis and back to waste) is shown with red arrows “open” and “close”. The amplitude of the reference signal is shown in current units nA (10^−9^ A). To facilitate reading, a magnified version of this figure is provided in Appendix A.

**Figure 2 ijms-23-04893-f002:**
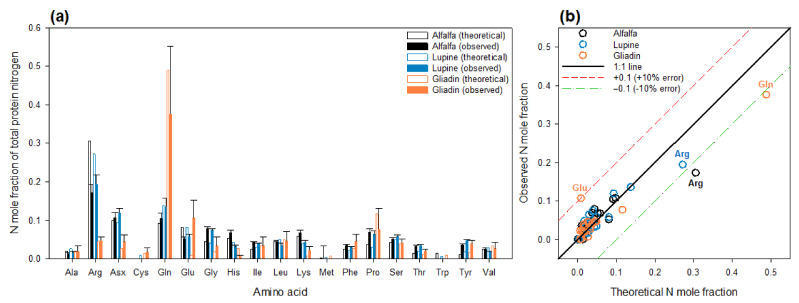
**Comparison of theoretical (from protein sequence) and observed (from mass-28 chromatogram) N mole fraction represented by amino acids.** (**a**) Individual comparisons between theory (empty bars) and observations (closed bars). (**b**) Graphical comparisons to show points that are close to the 1:1 line, except for Arg (lupine and alfalfa) and Gln (gliadin). The mole fraction calculated here is the fraction of N represented by the amino acid of interest with respect to total protein N. Due to the production of derivatives, corrections had to be made for some amino acids (see Section 4).

**Figure 3 ijms-23-04893-f003:**
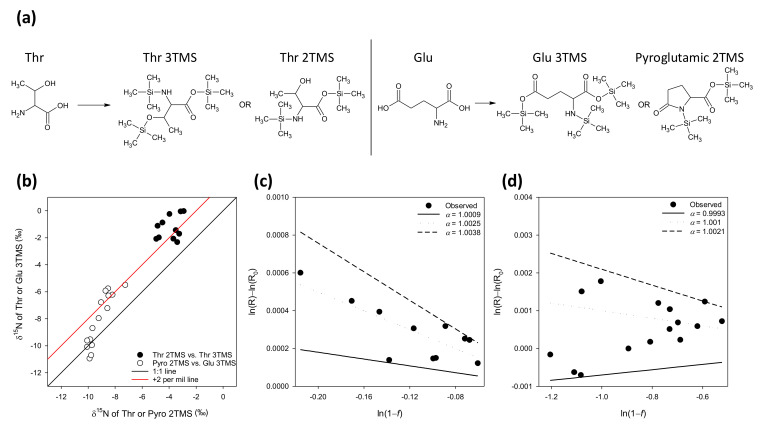
**^14^N/^15^N kinetic isotope effects in by-products formation during Thr and Glu derivatization.** (**a**) Product considered here: Thr with 2 or 3 TMS groups, and Glu 3TMS or pyroglutamate 2TMS production. (**b**) Relationship between δ^15^N values of products, showing the isotopic offset of roughly 2 per mil on average between products. (**c**) Calculation of the apparent isotope effect in “pyroglutamatization” of Glu 3TMS, using a log-log plot. (**d**) Calculation of the apparent isotope effect in the loss of a TMS group from Thr 3TMS, using a log-log plot (Rayleigh fractionation plot). The isotope effect for modelled lines (α) refers to the ^14^N/^15^N ratio of rates. The principle of Rayleigh’s equation is recalled in Section 4.

**Figure 4 ijms-23-04893-f004:**
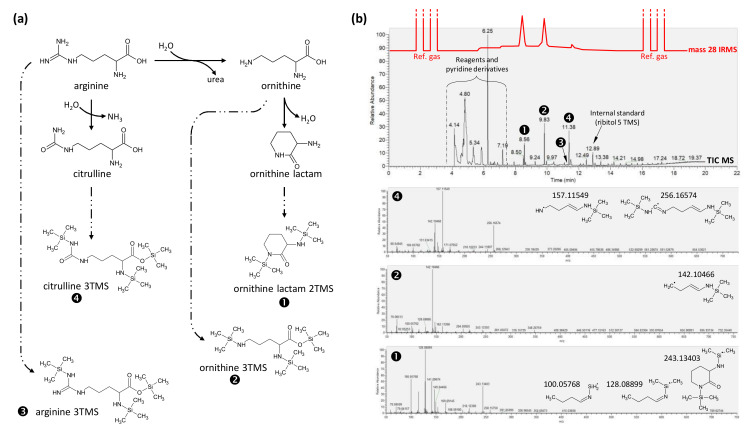
**Major products of arginine derivatization (silylation) under our conditions**. (**a**) Chemical pathway showing the four main products formed from arginine. (**b**) TIC trace of the GC-MS and associated MS spectra showing the three major peaks obtained: ornithine lactam 2TMS, ornithine 3TMS and citrulline 3TMS. Arginine 3TMS (peak no. 3) is of negligible importance (in contrast to derivatization carried out with methoxyamine, Appendix A). Major fragments and associated exact mass are shown on bottom panels. The IRMS trace (mass-28) is superimposed on top to show the modest contribution of citrulline (peak no. 4) in absolute terms (moles of N) compared to ornithine (peaks no. 1 and 2). To facilitate reading, a magnified version of this figure is provided in Appendix A.

**Table 1 ijms-23-04893-t001:** **δ^15^N values in selected amino acids, either pure standards or derivatives from protein hydrolysates**. When available, the comparison with the “true” value obtained by EA-IRMS is shown. The simple asterisk (*) indicates calculated values (weighted average, i.e., δ^15^N values weighted by relative mass-28 signal). The double asterisk (**) shows a common value for Ile, Pro and Gly derivatives in gliadin hydrolysates since it was difficult to do separate signal integrations to compute individual δ^15^N values. This table only shows δ^15^N values for amino acid derivatives with a sufficient signal to allow reliable determination (typically signal intensity >0.4 nA for mass-28; under this value, there is a significant loss in linearity). Results of statistical analyses (one-way ANOVA) are provided on the right-hand side, testing differences between GC-C-IRMS and EA-IRMS (pure amino acids), between species (protein hydrolysates) and between weighted average and EA-IRMS (protein isolates). Statistical classes are shown with lower case letters. NS, insignificant; S, significant (*p* < 0.05). Amino acids associated with potential methodological issues discussed in text (Section 2 and Section 3) are shown in *italics*, while others appear in **bold**.

	δ^15^N_GC-C-IRMS_			δ^15^N_EA-IRMS_	Statistics
*Pure amino acids*					
*Arg*	−5.31 ± 0.76			−5.51 ± 0.11	NS
*Glu*	−7.97 ± 0.80			−7.29 ± 0.11	NS
**Norleucine**	14.85 ± 0.30			14.02 ± 0.11	NS
*Thr*	0.54 ± 0.53			−0.19 ± 0.11	NS
*Protein hydrolysates*					
	*Gliadin*	*Lupine*	*Alfalfa*		
**Ala 2TMS**	n/a	−9.24 ± 2.56	−7.94 ± 2.07		NS
**Val 2TMS**	2.72 ± 2.13a	−3.91 ± 2.25b	−5.27 ± 0.20b		S
**Leu 2TMS**	0.45 ± 1.77a	−5.42 ± 1.19b	−3.78 ± 0.42b		S
*Ile 2TMS*	6.74 ± 1.05 **	−4.64 ± 1.66	1.31 ± 0.88		-
*Pro 2TMS*		1.5 ± 0.50	8.03 ± 0.70		-
*Gly 3TMS*		−3.28 ± 0.65	−0.94 ± 0.30		-
**Ser 3TMS**	0.55 ± 1.72	1.18 ± 2.65	0.99 ± 1.9		NS
*Thr 3TMS*	−2.67 ± 3.2a	14.06 ± 1.34b	14.68 ± 1.6b		S
*Ornithine lactam 2TMS*	3.20 ± 1.82a	0.68 ± 1.35a	−6.37 ± 1.25b		S
**Asp 3TMS**	−0.02 ± 2.60a	2.69 ± 0.28a	5.79 ± 0.05b		S
*Pyroglutamic 2TMS*	7.02 ± 1.26a	2.53 ± 0.36b	4.46 ± 0.75c		S
*Glu 3TMS*	5.82 ± 1.45a	2.48 ± 0.57b	2.90 ± 0.55b		S
**Phe 2TMS**	12.35 ± 2.75	8.70 ± 0.99	9.69 ± 1.30		NS
**Lys 4TMS**	1.10 ± 0.74a	−1.48 ± 0.50b	1.29 ± 0.08a		S
**Tyr 3TMS**	7.56 ± 2.33a	−1.4 ± 0.49b	0.31 ± 1.16b		S
*Protein isolates*					
Gliadin	4.86 ± 1.76 *			3.53 ± 0.05	NS
Lupine		0.57 ± 1.06 *		0.93 ± 0.05	NS
Alfalfa			2.14 ± 0.82 *	1.06 ± 0.04	NS

## Data Availability

Numerical raw data can be provided on request. All other data are presented in the paper.

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
