# Peer review of "Compound-Specific 14N/15N Analysis of Amino Acid Trimethylsilylated Derivatives from Plant Seed Proteins"

_ijms, 2022, doi:10.3390/ijms23094893_

Round 1

Reviewer 1 Report

Manuscript Compound-specific 14N/15N analysis of amino acids trimethylsilylated derivatives from plant seed proteins by
Jean-Baptiste Domergue, Julie Lalande, Cyril Abadie and Guillaume Tcherkez are designed according to the rules of the journal and contain all the necessary sections of the manuscript. The work leaves a good feeling of an honest, well-executed experimental-methodical scientific work. Some small remarks concern only the technical design of the article.
So, for example, in the introduction, the last paragraph contains tasks, but the overall clearly expressed goal is not formulated.
Figures 1,2,4 are not of very good quality - resolution and contrast need to be improved.
In figure 2b, it is impossible to disassemble something, it is better to separate it or make changes.
It is not clear what the porous line between the Thr and Glu modifications means, it is better to replace it with a solid line, the units of measurement are not obvious on the scales.
Currently, the drawings are regarded as an independent work and must contain complete information about the method and method of research. This is due to the fact that interdisciplinary work requires a more explicit and clear presentation of the results (or links as in Figure 2).
It would be better if table one would be formatted with Duncon's test and processed in ANOVA.
The description of the results contains multiple explanations that would be more appropriate in a discussion or methods section. In addition, I do not see a proposal to resolve a number of such contradictions, where the obtained value could be verified by introducing additional controls. Write it in the discussion, maybe as a section, or mark those data you are sure about in one color, and those that can be caused by sample preparation or other aspects of the method with another.
In general, the work is impressive and worthy, but it needs to improve the design and slightly edit the introduction and results.

Author Response

Jean-Baptiste Domergue, Julie Lalande, Cyril Abadie and Guillaume Tcherkez are designed according to the rules of the journal and contain all the necessary sections of the manuscript. The work leaves a good feeling of an honest, well-executed experimental-methodical scientific work. Some small remarks concern only the technical design of the article.

-Thank you for this positive comment.

So, for example, in the introduction, the last paragraph contains tasks, but the overall clearly expressed goal is not formulated.

-We have now added a sentence to clarify the overall goal of the paper.

Figures 1,2,4 are not of very good quality - resolution and contrast need to be improved.

-This is probably due to pdf conversion, since original figures are well-resolved and have a big size. When printed out on paper, these figures look good too. To avoid any problem, we have added magnified versions of figures 1 and 4 in Suppl Mat (figure 2 is at very high resolution).

In figure 2b, it is impossible to disassemble something, it is better to separate it or make changes.

-We suppose the referee refers to datapoint overlapping in panel 2b? If so, it is done on purpose, to show that all points fall in the same region, very close to the 1:1 line. Also, individual values can be found in panel a.

It is not clear what the porous line between the Thr and Glu modifications means, it is better to replace it with a solid line,

-We assume that the referee points to figure 3a. the porous line has been replaced by a solid line.

the units of measurement are not obvious on the scales.

-We have now added per mil (‰) on axes of figure 3b. In figures 3c and 3d, axes are dimensionless.

Currently, the drawings are regarded as an independent work and must contain complete information about the method and method of research. This is due to the fact that interdisciplinary work requires a more explicit and clear presentation of the results (or links as in Figure 2).

-We are unsure to understand what this comment refers to: is it chemical drawings of figure 3a? or the lack of information in the legend of Figure 3? To ensure figure 3 is clear enough, we have added explanation about log-log plots in the Material and Methods and referred to that in the legend.

It would be better if table one would be formatted with Duncon's test and processed in ANOVA.

-Statistics now done.

The description of the results contains multiple explanations that would be more appropriate in a discussion or methods section.

-It is true that the Results section contains explanations. We did this purposedly to ensure that the reader can follow easily. This is important we believe, because, as mentioned by the referee, this is an interdisciplinary work. Therefore, it is probably good to have simple explanations provided in Results. We also note that these pieces of explanation are not repeated in the Discussion, and thus it is OK from a formal point of view.

In addition, I do not see a proposal to resolve a number of such contradictions, where the obtained value could be verified by introducing additional controls. Write it in the discussion, maybe as a section, or mark those data you are sure about in one color, and those that can be caused by sample preparation or other aspects of the method with another.

-We are unsure to understand what the referee means by "contradictions". We assume that he/she refers to the fact that some isotopic values are more reliable than others due to the technical limitations mentioned in the Results and Discussion. We have adopted the suggestion of the referee to indicate this in Table 1, using bold and italic to indicate this.

In general, the work is impressive and worthy, but it needs to improve the design and slightly edit the introduction and results.

-We have improved the design as suggested (see above), and did amendments as requested.

Reviewer 2 Report

Compound-specific 14N/15N analysis of amino acid trimethylsilylated derivatives from plant seed proteins. This would allow the utilization of common protocols for metabolomics and isotopomics and thus facilitate metabolic studies. Overall, this technique may be suit- 23 able for compound-specific δ15N analysis, with pros and cons. The manuscript is properly written, and as a result, it may be approved for publication in its current form following a thorough review of its English language use.

Author Response

-Thanks for this positive comment. We have checked the English throughout and corrected small mistakes (in/during/within, was/were, etc.).

Reviewer 3 Report

The article is devoted to the isotopic analyses of nitrogen for evaluation of metabolic pathways of N nutrition. It is really very interesting to follow the way of N atoms during plants development. Especially it is concerned the differences between metabolomic pathways of serials and legumes, which were discussed in current investigation. Authors also described the difficulties and restrictions of such kind of investigation.

At the same time I have some remarks:

Materials used in the investigation are described too briefly. What kind of material it was? How it was prepared before chemical analyses?

As legumes absorb nitrogen from the air (via bacteria), what was the concentration of δ15N in the air?

The article can be published after some improvement.

Author Response

Materials used in the investigation are described too briefly. What kind of material it was? How it was prepared before chemical analyses?

-We are unsure to understand what this comment refers to, since the material is described in section 4.1, and protein purification and hydrolysis is explained in section 4.5. We have added some more information in section 4.1 to clarify the origin of products.

As legumes absorb nitrogen from the air (via bacteria), what was the concentration of δ15N in the air?

-By definition, d15N in air is zero per mil since atmospheric N2 is the international reference.

The article can be published after some improvement.

-Thanks for this comment.

Round 2

Reviewer 1 Report

The manuscript of Jean-Baptiste Domergue, Julie Lalande, Cyril Abadie and Guillaume Tcherkez was excellent even before the revisions. No further improvements are required and now it has become more clear and will be received with interest by specialists in this field. It must be published.